# Antibiotic Resistance Genes in Interconnected Surface Waters as Affected by Agricultural Activities

**DOI:** 10.3390/biom13020231

**Published:** 2023-01-24

**Authors:** Beilun Zhao, Peter M. van Bodegom, Krijn B. Trimbos

**Affiliations:** Department of Environmental Biology, Institute of Environmental Sciences, Leiden University, Einsteinweg 2, 2333 CC Leiden, The Netherlands

**Keywords:** environmental pollution, antibiotic resistance genes, agricultural area, pasture, greenhouse, co-occurrence, intl1 gene

## Abstract

Pastures have become one of the most important sources of antibiotic resistance genes (ARGs) pollution, bringing risks to human health through the environment and the food that is grown there. Another significant source of food production is greenhouse horticulture, which is typically located near pastures. Through waterways, pasture-originated ARGs may transfer to the food in greenhouses. However, how these pasture-originated ARGs spread to nearby waterways and greenhouses has been much less investigated, while this may pose risks to humans through agricultural products. We analyzed 29 ARGs related to the most used antibiotics in livestock in the Netherlands at 16 locations in an agricultural area, representing pastures, greenhouses and lakes. We found that ARGs were prevalent in all surface waters surrounding pastures and greenhouses and showed a similar composition, with sulfonamide ARGs being dominant. This indicates that both pastures and greenhouses cause antibiotic resistance pressures on neighboring waters. However, lower pressures were found in relatively larger and isolated lakes, suggesting that a larger water body or a non-agricultural green buffer zone could help reducing ARG impacts from agricultural areas. We also observed a positive relationship between the concentrations of the class 1 integron (intl1 gene)—used as a proxy for horizontal gene transfer—and ARG concentration and composition. This supports that horizontal gene transfer might play a role in dispersing ARGs through landscapes. In contrast, none of the measured four abiotic factors (phosphate, nitrate, pH and dissolved oxygen) showed any impact on ARG concentrations. ARGs from different classes co-occurred, suggesting simultaneous use of different antibiotics. Our findings help to understand the spatial patterns of ARGs, specifically the impacts of ARGs from pastures and greenhouses on each other and on nearby waterways. In this way, this study guides management aiming at reducing ARGs′ risk to human health from agricultural products.

## 1. Introduction

The discovery and application of antibiotics in the last century has largely facilitated medical treatments and improved agricultural practices and yields [1]. Many of these treatments are currently at risk [2,3,4] due to the recent and rapid global spreading of so-called antibiotic resistance genes (ARGs) [5,6]. ARGs are ancient and produced by microorganisms as a strategy to protect themselves from toxic compounds (e.g., antibiotics originating from other microorganisms) [7,8]. Bacteria may inherit or acquire ARGs by horizontal gene transfer [9,10]. Additionally, pathogenic bacteria that carry these ARGs might make the corresponding antibiotic ineffective [11,12,13]. Therefore, their rapid spread and occurrence in general and pathogenic bacteria poses a major problem to medical treatments. To date, ARGs have been detected in multiple industrial and agricultural areas [13,14,15]. If these ARGs are transferred to human pathogens, this could potentially cause serious global public health risks for humans [9,16,17,18,19].

Within agricultural areas, pastures have become one of the most important sources of ARGs, which is probably linked to the use of antibiotics in livestock [16,17,18]. To prevent health risks for livestock, antibiotics are commonly added to their diet, and the long-term exposure to antibiotics has led to a growing amount of ARGs in the intestinal flora of livestock [18,20]. For instance, tetracycline (49 tonnes, 31.8%) and the combination of sulfonamide and trimethoprim (30 tonnes, 19.5%) were the two most sold veterinary drugs in the Netherlands in 2020 (https://www.fidin.nl/, 1 June 2021). Correspondingly, high resistance to these three antibiotics was found in one important human pathogen *Salmonella typhimurium* isolated from cattle with 36.4% resistance to tetracycline and 45.5% to sulfonamide, and from humans with 33.3% resistance to tetracycline and 34.6% to sulfonamide [21]. This supports the transfer of ARGs from agricultural products (e.g., meat and milk) [22,23,24] to human bodies [22,23,24,25], and the risks of ARGs to human health. At present, a different cluster of antibiotics is used for humans from livestock to minimize the risk of transferring ARGs from pasture products to human medicine [21]. However, pasture-originated ARGs may hinder potential antibiotic repurposing in human medicine, which negatively affect strategies fighting the potential emergence of new resistance mechanisms [26,27,28].

ARGs from the livestock microbiome are continuously released into the environment through their feces, thereby polluting adjacent soil and water [29], and through dispersal by airborne release [30,31]. Some of the ARGs released to the environment end up in waterways [32], which first and foremost causes risks to natural ecosystems [17,30,33]. Another potential risk to human health arises if they enter greenhouse food production cycle through the water that is used for irrigation in horticulture, which may subsequently cause ARGs to accumulate in vegetables [34]. In areas such as the Netherlands, the close proximity of intensively used pastures with livestock and horticulture in combination with the omnipresence of waterways potentially facilitates this pathway of ARGs and its transfer to humans through diet [17]. However, to what extent waterways can bring pasture-originated ARGs to neighboring horticultural regions and how to weaken this transfer by geographic planning is not clear. A deep understanding of ARG prevalence and distribution is therefore important to develop strategies and policies in agricultural area management aimed to reduce the risks of ARGs to human health.

Another element to consider in public health strategies is the potential co-occurrence of ARGs. Current policies aim to reduce the combined effects of ARGs from agriculture on human health by using different antibiotics in animals compared to human medical treatments. For example, in the Netherlands, penicillin is primarily used in human health care, while trimethoprim and sulfonamides are mainly used in livestock [21]. This strategy combined with the controlled use of antibiotics in public health very likely contributed to the lower antibiotic resistance rates in the Netherlands compared to other European countries [35]. However, this strategy may be invalid due to the co-occurrence of ARGs related to different antibiotics in the environment [15,36,37]. Yet inconsistent co-occurrence patterns between ARGs have been observed [38,39], which makes our understanding of how ARGs co-occur incomplete. For instance, the co-persistence of a sulfonamide and tetracycline ARG was observed in deep layer sediments but not in surface sediments or water [38], while a co-occurrence of these compounds was shown in water yet changed with time in another study [39]. Thus, to guide ARG management in agricultural areas, more knowledge of the co-occurrence of ARGs and how environmental drivers influence this co-occurrence is urgently needed, especially for densely populated agricultural areas.

Horizontal transfer additionally leads to the dispersion and increase in ARGs. The class 1 integron (intl1) gene is one of the most important horizontal transfer genes [14,40]; however, its connection with ARGs′ prevalence seems context dependent [38]. Several abiotic factors have been shown to affect this connection and impact the prevalence of ARGs [41,42]. For example, pH plays an important role in regulating the horizontal transfer and assembly process of ARGs [9,37,40,43], and dissolved oxygen (DO) was correlated to ARGs′ abundance [13]. However, these effects were only found across large environmental gradients or between different types of samples. To further understand the role of abiotic factors on the dispersion and increase in ARGs in agricultural environments, their impacts on horizontal transfer genes and the prevalence of ARGs are to be included in ARG assessments.

The information on ARGs′ prevalence and distribution in the surface freshwater system in agricultural areas is therefore essential to guide measures aiming at reducing the risk of ARGs originating from the food production chain to affect human health. The co-occurrences and differences in ARGs′ prevalence between different sites will help us understand how to reduce the spread of ARGs from pastures to neighboring areas. Furthermore, the co-occurrence of ARGs will illuminate the mechanism behind the prevalence of ARGs, and may guide choices of antibiotic usage [12,44]. Moreover, understanding how biotic and abiotic elements are involved in the presence of ARGs will help to determine whether controlling certain factors can reduce the spread of ARGs [9,40].

The present study assesses the possible impacts of pastures on the prevalence and composition of ARGs in nearby surface waters, including the ditches around greenhouses and large water bodies (lakes). Water samples were collected from a total of 16 locations, including six ditches around pastures, six ditches around greenhouses and four lakes in the same area. The relative concentrations of 29 ARGs that indicate resistance to the four classes of most used antibiotics in the Netherlands livestock and multidrug-resisting ARGs were quantified to analyze their distribution in this pasture–greenhouse area. The relationships between the prevalence of these ARGs were analyzed, while accounting for the potential influences of horizontal gene transfer and four environmental factors (phosphate, nitrate, pH and DO) on the prevalence of ARGs.

## 2. Materials and Methods

### 2.1. ARG Choices

Over the last decade, sulfonamide, trimethoprim, tetracycline and beta-lactamase antibiotics have been the most commonly sold and used antibiotics in livestock in the Netherlands [21]. The resistance against these four antibiotics is also widely prevalent in Europe. For example, sul1 and tetM genes resisting sulfonamide and tetracycline separately have been found across the Rhine River. Therefore, we selected ARGs that are related to these antibiotics and that are known to prevail in Europe [9,19,33,45,46,47]. Additionally, we selected the multidrug ARGs. The multidrug ARGs help bacteria to move antibiotics out of the cell and are widely spread and were therefore included [5,48]. A total of 29 ARGs falling in five antibiotic classes (Table 1) were included and quantified in this study. To quantify the potential impacts of horizontal gene transfer on the abundance and co-occurrence of ARGs, the concentration of intl1 was also determined.

To further understand the mechanisms of action of the ARGs and their co-occurrences, these ARGs are also classified into three categories based on their mode of action, including protection (sul1, sul2, sul3, tet36, tetM, tetS), deactivation (ampC, blaCMY-2, blaNPS-2, blaOXA-1, blaSHV34, blaTEM1, blaZ, dfr13, dfrA19, dfrll, dfrV, dhfr1, imp-2, tetX) and efflux (qacE∆1, acrB, acrD, mexD, mexl, qacF, tetA, tetB, tetK).

### 2.2. Sampling Locations and Collection

Water samples were collected from 16 locations at Delfgauw, the Netherlands (Figure 1) in the period of May–June 2021. Greenhouses are located in the central area, while pastures are in the northern and southern parts of the area. All greenhouses and pastures are surrounded by ditches or small canals and connected with each other by waterways. Several lakes are located in the corners of the area. The sampled water bodies are unlikely to have received ARGs from outside the agricultural areas because there are hardly any domestic or industrial sewage effluents in this area. Water samples were collected from six locations within the greenhouse area, six locations in pasture areas and four lakes. Lakes were included to explore how pastures influence nearby big water bodies with respect to ARGs. At each location, surface water samples were collected from three sub-sites at an interval of 5–10 m to better represent the location. Geographic locations of all sampling sub-sites are shown in Table A1. These sub-samples were mixed and filtered as one water sample representing the location. First, each water sample was pre-filtered using a plastic syringe (BD Plastipak™) and pushing it through a 5 μm polyethersulfone (PES) membrane filter in a membrane container, to remove the large-sized materials while keeping the bacteria, allowing more water to pass the bacteria-catching filter. Next, around 20–180 mL of pre-filtered water was filtered using a 0.2 μm PES membrane filter to catch bacterial DNA. The filtration was repeated once more to collect more bacterial DNA. Each filter was immediately put into a 2 mL tube with 700 μL CTAB Lysis buffer (AppliChem GmbH, DE) and stored at 4 °C. To avoid contamination, all syringes, membrane containers, and glassware were soaked in a 10% bleach solution over 10 min before being washed by deionized water, then air-dried on clean paper towels before being used [49]. Water sample collection from each location was repeated three times, in one-week intervals, to avoid short-term variations in ARG concentrations. This resulted in a total of 48 samples for further ARG analysis. Two samples of 180 mL Milli-Q water were filtered following the same protocol and analyzed for ARGs to explore whether ARG contamination took place during the filtration procedure.

### 2.3. DNA Extractions and ARGs Quantification

One day after filtration, DNA was extracted from each 0.2 μm PES membrane following a CTAB protocol as used in several previous studies [50,51,52] before eluting in 100 μL Tris-EDTA buffer solution (Sigma-Aldrich, St. Louis, MO, USA). The extracted DNA from the two 0.2 μm PES membranes of one water sample was combined to reach an end volume of 200 μL. All extracted DNA samples were stored at −20 °C till further steps within one month. 

All DNA samples including the two control samples were subsequently analyzed using the QX200 ddPCR system (Bio-Rad). The absolute copy concentration (copies/μL) of each of the 29 ARGs in each DNA samples was quantified using an EvaGreen assay, and the sequence and annealing temperature of each pair of primers are shown in Table A2. The intl1 and 16S rRNA gene copy concentrations were quantified using a probe assay. Primer sets and probes (Table A3) were selected from previous studies and ordered from Sigma-Aldrich (Darmstadt, Germany). The EvaGreen reaction mixes were set to a total of 20 μL including 2 μL DNA template, 250 nM of each forward and reverse primers, 10 μL QX200™ ddPCR™ EvaGreen Supermix and DNase/RNase-Free Distilled Water (Invitrogen™ UltraPure™, Thermo Fisher Scientific, New York, NY, USA). The DNA template for the probe reaction was diluted 10^3^ times for 16S rRNA gene quantification because its concentrations were beyond the ddPCR range (<2000 copies/μL). The probe reaction mixes were performed on a total of 20 μL including 2 μL DNA template, 900 nM of each forward and reverse primers, 250 nM of TaqMan probe and 10 μL ddPCR™ Supermix for probes (No dUTP) and DNase/RNase-Free Distilled Water (Invitrogen™ UltraPure™, Thermo Fisher Scientific, New York, NY, USA). Every sample was run in duplicate. Each plate contained two negative controls using Tris-EDTA buffer solution as a template. The EvaGreen assay thermal reactions were carried out as follows: 5 min at 95 °C, 40 cycles of 30 s at 95 °C and 1 min at the annealing temperature of the corresponding primers (Table A2), then 5 min at 4 °C, 5 min at 90 °C before 4 °C conservation. The probe assay thermal reactions were as follows: 10 min at 95 °C, 40 cycles of 30 s at 94 °C, and 1 min at the annealing temperature of corresponding primers (Table A3), then 10 min at 98 °C before a hold at 4 °C. The concentration in copies/μL of each sample was calculated as described by 48 through merging the duplicate measurements using QX200 Droplet Reader and QuantaSoft (V.1.7.4, Bio-Rad Laboratories B.V., Veenendaal, NL) following the ddPCR protocol. The ddPCR results of all ARGs and the intl1 gene were converted into relative concentrations (copies/16S rRNA gene) and absolute concentrations (copies/mL water sample) based on the various dilutions involved and the amount of filtered water in the analysis process. No ARG or intl1 gene was found in any of the controls nor the ddPCR blanks.

### 2.4. Environmental Factors (Phosphate, Nitrate, pH and DO)

During the water sampling, the DO and pH of the water at each location were measured, utilizing HANNA-EDGE instruments with a DO probe (HI764080) and a pH probe (HI-11310), respectively. Additionally, phosphate and nitrate concentrations were quantified, because they are usually high in agricultural areas due to the application of fertilizers and may covary with the prevalence of ARGs. Four tubes of 50 mL water were obtained from the same location and at the same time, using 50 mL sterilized centrifuge tubes from SARSTEDT (Nümbrecht, Germany), resulting in a total of 192 tubes of water transferred to the lab for quantification of phosphate and nitrate concentrations within several hours. The quantifications were carried out using the MERCK Phosphate Test kit (product No. 114848, Merck KGaA, Darmstadt, Germany) and the MERCK Nitrate Test kit (product No. 109713, Merck KGaA, Darmstadt, Germany) following the manufacturer′s protocols.

### 2.5. Statistical Analysis

R software (V.4.1.1) was used throughout the data analysis process. ggplot2 (V.3.3.5) and ggpuber (V.0.4.0) packages were used to generate figures. To evaluate differences in the total concentration of ARGs (both relative and absolute) between land use types, a Kruskal–Wallis test was used. Kruskal–Wallis tests were also run to evaluate differences in the total relative concentration of the different ARG classes and mechanisms. Subsequently, a PERMANOVA was performed to analyze the ARG differences (per ARG, class and mechanism separately) between land use types. To further explore which ARG differ between each pair of land use types, a Dunn′s test was applied using PMCMRplus package (V.1.9.3). Non-metric multidimensional scaling (NMDS) analysis and multivariate homogeneity of group dispersions utilizing the Vegan package were used to further analyze the differences within land use types. Additionally, to explore the differences between the 16 locations, another PERMANOVA was used to analyze the composition differences (for each ARG, class and mechanism separately) between individual locations, followed by a pair-wise comparison (only for ARGs composition) using the Vegan package. Due to the limited number of replications at each location, *p* = 0.1 was used as significance threshold. 

A final PERMANOVA was performed to explore the impacts of the four abiotic factors including phosphate, nitrate, pH and DO, and the intl1 gene on ARG composition (for each ARG, class and mechanism separately). Linear models in R were used to further analyze the connection between the intl1 concentration and the total relative concentration of ARGs, and between the intl1 concentration and each ARG class and mechanism. Spearman′s rank correlation coefficients and the significance were calculated between ARG classes and mechanisms separately using the Formula package (V.1.2-4).

## 3. Results

### 3.1. Differences between Land Use Types

All 29 ARGs and the intl1 gene were found in all locations (i.e., in at least one of the three replicates of each location) except for blaNPS-2 which was absent in all three samples from one lake location (lake 3). This ARG was found in the least number of samples (34 of the 48 samples), followed by the dfrA19 being detected in 38 samples. Next were tetS and blaSHV34 which were detected in 41 and 42 samples, respectively. AmpC, mexD, tetX, tetK, tetM, dfr13 and dfrV were absent in no more than three samples. The intl1 gene was detected in all samples except for one sample from lake 4. Overall, the three sulfonamide ARGs (sul1, sul2, sul3) dominated the ARGs in all locations except for lake 4 (Figure 2a,b), while other ARGs were present in very low percentages.

To explore the differences in ARG prevalence between land use types, both the total concentration (both relative and absolute) and the composition of all ARGs were analyzed. The total absolute concentration of all ARGs was much lower in lakes than in pastures and greenhouses, while the total relative concentration of all ARGs did not show significant differences between land use types (Figure 3a,b, Table 1). Interestingly, the ARG composition calculated using the relative concentration of each ARG was different between land use types (PERMANOVA, *p* < 0.01). The Dunn′s test showed that two ARGs were significantly different between greenhouses and lakes (*p* < 0.05), including one sulfonamide ARG (sul 2) and one tetracycline ARG (tetK) (Table 1). Six ARGs showed significant differences between pastures and lakes (*p* < 0.05), including four tetracycline ARGs (tetA, tetX, tetK and tetS), one beta-lactamase ARG (imp-2) and one multidrug ARG (acrD). After clustering the ARGs according to their antibiotic class and mechanism, the ARG composition also showed strong differences between land use types (PERMANOVA, *p* < 0.01). Two classes and two mechanisms showed no differences between land use types (Figure 4). While no individual trimethoprim ARG had shown significant differences, trimethoprim ARGs as a group showed significant differences between greenhouses and lakes, and between pastures and lakes. Further analysis with an NMDS, however, showed that the ARG composition of lake 4 was largely different from all other locations, indicating the ARG differences between land use types might have originated from the deviating behavior of lake 4 (Figure 4c). The analysis of the beta dispersion confirmed a significant difference in variance between land use types, likely driven by lakes. After removing lake 4 from the analysis, the PERMANOVA indicated no differences in ARG composition between land use types (nor in individual ARG, class or mechanism).

When analyzing differences among all 16 individual locations (as factor), the compositions of individual ARGs, and when clustered by antibiotic class and mechanism were all significantly different between locations (also see Figure 4c) no matter whether including lake 4 or not (Table 2). The subsequent pair-wise analysis on the ARG composition between locations indicated (Table A4) that the six greenhouse locations were internally homogeneous. In contrast, the pastures fell into two groups, with pastures 1–3 and 6 being similar to each other and different from pastures 4 and 5. Lakes 1 and 2 were quite similar to greenhouses and pastures, while lakes 3 and 4 were different from almost all other locations. In combination, this suggests that ARG composition was affected by both differences between land use types, as well as between locations, with lakes 3 and 4 showing the most deviating composition in ARGs.

### 3.2. The Influences of Abiotic Factors and Intl1

Phosphate, nitrate, pH and DO of the sampled surface freshwater in this study did not show much variation with 0.01–3.2 mg/L, 0.6–6.9 mg/L, 7.3–9.3 and 3.1–9.8 mg/L, respectively. These four abiotic factors showed no impacts on the ARG composition (*p* > 0.05, per ARG, class and mechanism separately), as indicated by the PERMANOVAs. However, the PERMANOVAs showed a significant impact of the intl1 gene concentration on ARG composition (*p* < 0.01). To further analyze the intl1 impact on each antibiotic class and mechanism, linear models were used. The results show a strong relationship between intl1 concentrations and the occurrence of trimethoprim, sulfonamide and multidrug classes, and the protection mechanism (Table 3, Figure 5 and Figure 6). Additionally, the total relative concentration of ARGs was significantly related to intl1 concentration (*p* < 0.01).

### 3.3. Co-Occurrences of ARGs

Spearman′s rank correlation coefficients showed positive correlations between all pairs of ARG classes, except for between sulfonamide and three other classes (tetracycline, beta-lactamase and multidrug), and a negative correlation between sulfonamide and trimethoprim (Table A5). Correlation coefficients between the relative concentrations of ARGs for each mechanism were only significantly positive between deactivate and efflux (R = 0.6, *p* < 0.01) (Table A6).

## 4. Discussion

The aim of this study was to assess the potential impact of ARGs originating from pastures on nearby waterways and to suggest ways to reduce this impact. Therefore, we analyzed 29 ARGs related to the most commonly used antibiotics for livestock in the Netherlands at 16 locations in an agriculture area, representing pastures, greenhouses and lakes. The concentration of the intl1 gene—as a metric of horizontal gene transfer—and four abiotic factors (phosphate, nitrate, pH and DO) were investigated to understand factors associated with ARGs prevalence. Co-occurrences between ARGs were analyzed to recognize their relations. The results showed that ARGs were prevalent in all locations. ARG composition was similar between land use types except for a deviation caused by lake 4. Differences in ARG composition within land use type were found in pastures and lakes, while the composition was similar within greenhouses. None of the abiotic factors tested significantly impacted ARG composition. The intl1 gene concentration was related to ARGs concentration and composition, which supports the potential importance of horizontal gene transfer in the dispersal of ARGs. ARGs of different classes and from different mechanisms were shown to co-occur, possibly induced by simultaneous use of different antibiotics.

The 29 ARGs and intl1 gene were present in all pasture ditch locations in this study. These ARGs could potentially bring antibiotic resistance risk to nearby areas and humans through the various pathways outlined in the introduction section [12,15,30,45,53]. For example, these ARGs could pollute vegetables in the greenhouses close to the pastures if the nearby water is used for irrigation. Sulfonamide ARGs were most prevalent in the pastures which corresponds to the usage of sulfonamide in livestock in the Netherlands [21]. However, while similar amounts of trimethoprim, tetracycline and beta-lactamase are used compared to sulfonamide [21], the concentrations of ARGs resistant to these antibiotics were much lower. One possible explanation is that sulfonamide is used more frequently in the pastures of the studied area than on average in the Netherlands. Another potential reason is that the higher diversity of ARGs in the antibiotic classes of trimethoprim, tetracycline and beta-lactamase, compared to sulfonamide, causes a higher variety of ARGs in the pasture ditches [33,47,54,55]. Some ARGs in these three antibiotic classes may not have been part of the analysis in this study.

Unexpectedly, the ditch locations surrounding greenhouses demonstrated similar ARG concentrations and composition as the ditch locations around pastures. Locations close to both greenhouses and pastures contained even higher ARG concentrations, indicating that next to pastures, greenhouses might also act as an ARG source thereby pressuring nearby areas. ARGs have been widely observed in greenhouses in both soil and plants and their occurrence has been suggested to also be induced by pesticides, fertilizers and other stressors [56,57,58]. Pesticides can select for ARGs in addition to antibiotics, and the accumulation of heavy metals in soils under the influence of pesticides possibly causes an increase in ARGs [58,59]. Additionally, heavy metals have been confirmed to positively correlate to ARGs [13,15,42,60]. In this study, we only tested four ARG classes related to antibiotics used in livestock and multidrug ARGs. If we would have additionally tested ARG classes related to activities in greenhouses, e.g., ARGs resisting aminoglycoside and glycopeptide antibiotics [61], the result might have shown more diverse and higher amounts of ARGs around greenhouses. Consequently, the greenhouse is highly likely to be another important ARG source in addition to pastures. The ARGs from pastures and horticulture also seemed to impact ARGs in neighboring lakes, while this impact was less pronounced in relatively larger and isolated lakes. Lakes 3 and 4 were 5–8 times larger than lakes 1 and 2, and were surrounded by woods (Figure 1) and contained lower concentrations of ARGs. Moreover, lake 4 contained a different ARG composition compared to all other locations. This indicates that a larger water area or a non-agricultural green buffer zone (e.g., forest) may help to reduce ARGs′ impact from pastures and horticulture on neighboring areas. Moreover, considering sufficient distance and non-agricultural green space between pastures and greenhouses is essential in the context of aggravated ARG risks to human health through diets (agricultural products).

The four abiotic factors tested in this study showed no impacts on ARGs in these agricultural water bodies, indicating weak correlations, or complex co-effects with other unexplored factors. While pH was previously found to be an important condition to the horizontal transfer and assembly process of ARGs thus influencing the prevalence of ARGs within a tested pH ranged from 4 to 10 [9,37,40,43], it was demonstrated not to influence ARGs in this study. These contrasting results might be explained by the narrow range of pH (7.3–9.3) of surface waters in this study. The prevalence of ARGs has been reported to vary with DO between soil, freshwater and gut samples [13], while DO did not correlate to ARGs′ presence in this study. The similarity in chemical composition of the surface water in the present study may explain this absence of effects and also why phosphate and nitrate had no impacts on ARGs. An alternative explanation is the combined effects of multiple factors on the distribution and prevalence of ARGs in a field situation, where factors are difficult to impose or separate compared to a laboratory study. Overall, the four abiotic factors were not suitable as the indicators of ARGs′ prevalence assessment, and reducing ARGs is not possible by controlling the values of these factors in agricultural water bodies.

Horizontal gene transfer may also have reduced the impacts of the abiotic factors tested [9]. Integrons are an important form of mobile gene elements that can promote ARG horizontal transfer among bacteria, thereby contributing to ARGs increase and bringing risk to human beings [19]. Intl1 is the predominant integron gene contributing to the ARGs prevalence in bacterial communities, and it is an indicator of the process of horizontal gene transfer [62]. Previous studies showed strong correlations between the intl1 gene and ARGs resisting several common antibiotics, such as tetracycline and sulfonamides [14,40,44,45,63,64]. Yet, this correlation is increasingly demonstrated to be complex [58,65]. To quantify potential impacts of horizontal gene transfer on the abundance and co-occurrence of ARGs in freshwater systems in agricultural areas, the concentration of intl1 was determined. Our results support the role of horizontal gene transfer—indicated by the abundance of the intl1 gene—on both the abundance and composition of ARGs in agricultural areas. However, the correlation between the intl1 gene and ARG classes varied, and was either positive (sulfonamide and multidrug), negative (trimethoprim) or non-related (tetracycline and beta-lactamase). The intl1 gene showed a positive correlation with protection mechanism ARGs rather than deactivated and efflux mechanisms, which is possibly explained by the dominance of the intl1-positively correlated sulfonamide ARGs in this mechanism group. Additionally, another study indicated a correlation between intl1 and overall ARG concentrations, induced by a correlation with sul1, under a similar situation that sul1 dominated the ARGs [45]. The consistent connection between ARG concentrations and intl1 suggests that horizontal gene transfer is an important contributor to the distribution of ARGs in the environment and especially so for sulfonamide ARGs, which seems particularly susceptible to horizontal gene transfer based on these observations. Sulfonamide ARGs have also been found to be the dominant ARG in many other studies [45,66,67]. We also found its dominance throughout the agricultural area excluding one distant lake. However, sulfonamide ARGs cannot be indicators of total ARGs prevalence because it was not positively correlated to other ARG classes.

Insight into the co-occurrences of ARGs is essential to ARG assessments and contributes to formulating strategies in reducing ARG risks. For example, the co-occurrences of ARGs resistant to tetracycline and beta-lactamase identified in the present study possibly indicate that applying these two antibiotics in livestock and human separately will not prevent the pasture-originated beta-lactamase resistance risk (induced by the tetracycline ARG co-occurrences) to human health. However, compared to other locations closer to agriculture areas, lake 4 contained low concentrations and a different composition of ARGs. The ARGs in lake 4 may have come from humans since it is commonly used for recreational swimming. This different ARG composition compared to agricultural areas supports that applying different antibiotics to livestock and human helps to prevent pasture-originated ARGs to humans. Therefore, the co-occurrences of ARGs were very likely caused by the synchronous applications of different antibiotics instead of by intimate connections between ARG classes. Co-occurrence was also found between efflux and deactivated mechanisms, indicating synergies between ARGs. This may again be due to the co-occurrences of ARG classes and hence the co-occurring application or distribution of ARGs, since the ARGs in these two mechanism groups correspond to four ARG classes which were all positively connected with each other. The efflux mechanism group contained all ARGs in the multidrug class and three ARGs from the tetracycline class, and the deactivate mechanism group contained all ARGs in the beta-lactamase and trimethoprim classes and one in the tetracycline class. In contrast, sulfonamide and trimethoprim are usually applied in combination [68], while a negative correlation between ARGs resisting these two antibiotics was observed. Overall, the exact causes of co-occurrences of ARGs are still unclear and require additional research.

Based on the distribution and prevalence patterns of ARGs obtained from this study, future studies might be designed to understand and predict the mechanisms of co-occurrences and transfer of ARGs. Measurements on the prevalence of bacterial strains and associated conjugation experiments can help understand the connection between ARGs prevalence and bacteria. Such measurements may also help in understanding mechanisms of ARG transfer, since bacteria have been found to play an important role in the spread of ARGs though horizontal transfer [12,18,69].

## 5. Conclusions

In conclusion, ARGs are prevalent in a low-lying agricultural area in the Netherlands connected through waterways. Both pastures and horticulture generate ARG pressures on nearby waterways and impact each other, consequently causing ARG pollution to the environment and may spread ARG risks to humans. Our data suggest that horizontal gene transfer, next to physical transport, strongly contributes to the widespread distribution of ARGs, particularly so for sulfonamide ARGs. Fortunately, geographic isolation can help reduce the impact of ARGs from agricultural areas. Hydrological isolating pastures and greenhouses can possibly contribute to weakening their crossover ARG impacts. Co-occurrences between ARGs depend on both the class and mechanism and are likely caused by the simultaneous use of different antibiotics. These ARGs originating from agricultural areas will possibly be transported to distant humans by the transportation of agricultural products (animal food products, vegetables and flowers), and cause human disease and the invalidation of antibiotics, which may result in death in some cases.

## Figures and Tables

**Figure 1 biomolecules-13-00231-f001:**
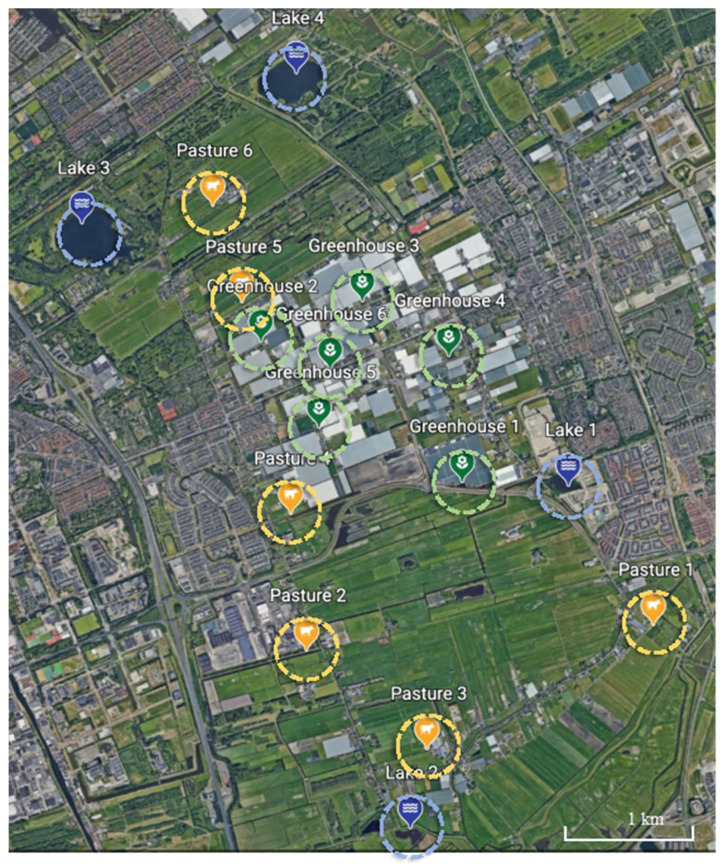
Sampling locations. The colors indicate the location types, including pasture (yellow), greenhouse (green) and lake (blue). Each location is highlighted by a circle with a diameter of 500 m to indicate the zone of immediate influence on the sampling location.

**Figure 2 biomolecules-13-00231-f002:**
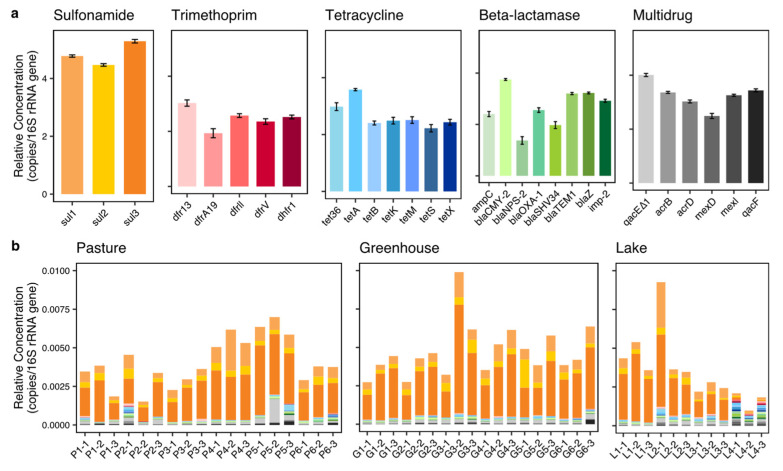
ARG concentrations. (**a**) Average of the relative concentrations [log^10^(x*10^8^+1) transformed, copies/16S rRNA gene] of each ARG in all 48 samples, with error bars on the top showing the standard errors (SE). Colors indicate ARGs, each color family represents an antibiotic class, sulfonamide (yellow), trimethoprim (red), tetracycline (blue), beta-lactamase (green) and multidrug (grey). (**b**) Relative ARG concentrations (copies/16S rRNA gene) of each sample. Row names indicate the land use type of the sampling location (P, pasture; G, greenhouse; L, lake) followed by the number of the sequence of locations and replicates, e.g., P1 (the first pasture location)-1 (the first sampling replication). Colors indicate ARGs in the same way as in (**a**).

**Figure 3 biomolecules-13-00231-f003:**
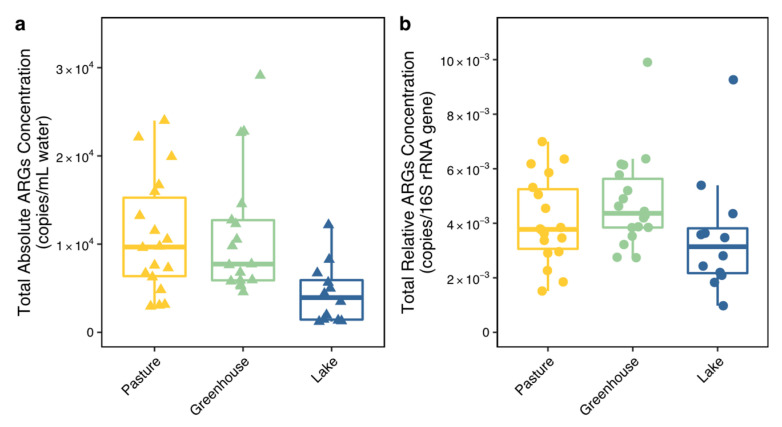
Total absolute ARG concentrations (copies/mL water) (**a**) and total relative ARG concentrations (copies/16S rRNA gene copy) (**b**). Colors indicate the location types, including pasture (yellow), greenhouse (green) and lake (blue).

**Figure 4 biomolecules-13-00231-f004:**
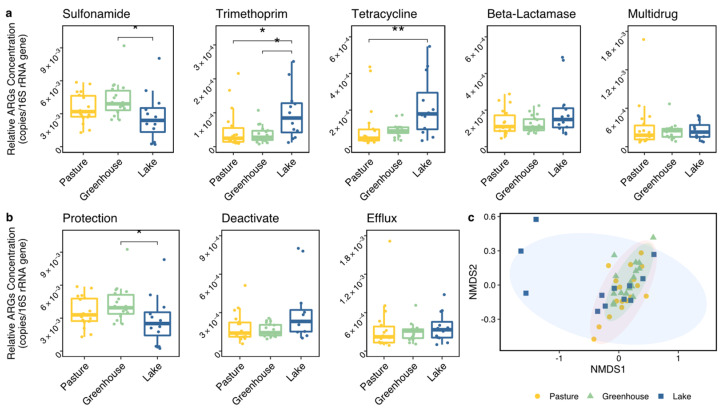
(**a**) The total relative ARG concentration of each antibiotic class (copies/16S rRNA gene), including sulfonamide, trimethoprim, tetracycline, beta-lactamase and multidrug. (**b**) The total relative ARG concentration of each antibiotic mechanism (copies/16S rRNA gene), including protection, deactivate and efflux. Significant differences between land use types from the Dunn′s test as the post-hoc test of a Kruskal–Wallis test are shown by asterisks on the top. Note that the ranges of the different y-axes are different. (*, *p* < 0.05; **, *p* < 0.01). (c) NMDs diagram of land use types based on the ARGs composition. Colors and shapes indicate the land use types.

**Figure 5 biomolecules-13-00231-f005:**
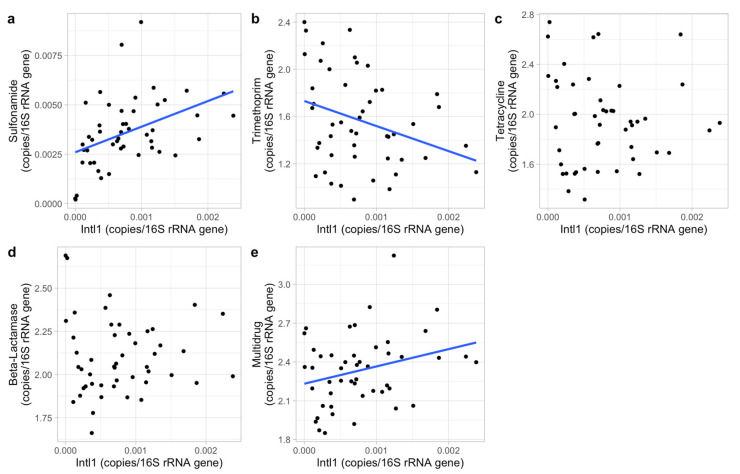
The relative concentration of the intl1 gene (copies/16S rRNA gene) for each ARG class ((**a**), sulfonamide; (**b**), trimethoprim; (**c**), tetracycline; (**d**), beta-lactamase; (**e**), multidrug). The ARG concentration was log^10^(x*10^6^) transformed except for sulfonamide to meet normal distributions. Significant relations are shown by the blue lines indicating the linear regression line for each data set.

**Figure 6 biomolecules-13-00231-f006:**
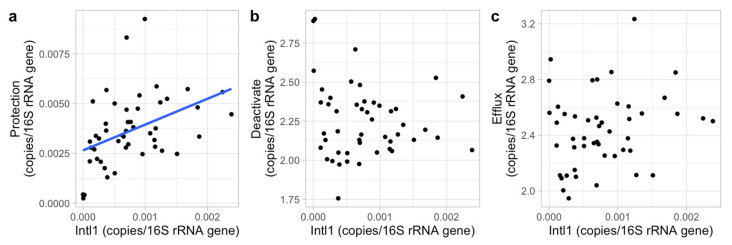
The relative concentration of the intl1 gene (copies/16S rRNA gene) for each ARG mechanism ((**a**), deactivate; (**b**), protection; (**c**), efflux). The ARG concentration was log^10^(x*10^6^) transformed except for protection to meet normal distributions. Significant relations are shown by the blue line indicating the linear regression line.

**Table 1 biomolecules-13-00231-t001:** *p* values of pairwise comparisons using Dunn′s test as the post hoc test of the Kruskal–Wallis test for evaluating differences between each pair of land use types. The absolute and relative concentrations of all ARGs, and the relative concentration of each class of ARGs and each ARG were included. *p* values were corrected for type I errors using Holm.

	ARGs	Greenhouse and Pasture	Greenhouse and Lake	Pasture and Lake
Total absolute ARG concentrations		0.703	0.003 **	0.006 **
Total relative ARG concentrations		0.35	0.05	0.35
Beta-lactamase		0.91	0.53	0.53
	ampC	0.57	0.14	0.28
	blaCMY-2	1	1	1
	BlaNPS-2	1	1	1
	blaOXA-1	0.65	0.48	0.65
	blaSHV34	1	1	1
	blaTEM1	0.73	0.73	0.39
	blaZ	1	1	1
	imp-2	0.227	0.227	0.024 *
Multidrug		1	1	1
	qacE∆1	1	1	1
	acrB	1	1	1
	acrD	0.439	0.156	0.042 *
	mexD	0.82	0.86	0.86
	mexl	1	1	1
	qacF	0.88	0.11	0.11
Sulfonamide		0.342	0.032 *	0.342
	sul1	0.668	0.068	0.116
	sul2	0.2482	0.0044 **	0.0638
	sul3	0.313	0.086	0.357
Tetracycline		0.1507	0.1507	0.0063 **
	tetA	0.91	0.1599	0.0042 **
	tetX	0.1934	0.1934	0.0075 **
	tetB	0.9	0.87	0.87
	tetK	0.803	0.028 *	0.035 *
	tetS	0.634	0.06	0.028 *
	tet36	0.53	0.53	0.18
	tetM	0.91	0.58	0.58
Trimethoprim		0.868	0.042 *	0.042 *
	dhfr1	0.87	0.38	0.38
	dfrll	0.81	0.81	0.62
	dfr13	0.56	0.49	0.27
	dfrA19	0.41	0.49	0.21
	dfrV	0.99	0.73	0.99

* indicates *p* < 0.05, ** indicates *p* < 0.01.

**Table 2 biomolecules-13-00231-t002:** *p* values of a PERMANOVA test between land use types and individual locations for individual ARGs, class and mechanism composition.

	ARGs	Class	Mechanism
Land use type	0.002 **	0.008 **	0.002 **
Land use type (without lake 4)	0.29	0.243	0.312
Location	0.001 **	0.001 **	0.001 **
Location (without lake 4)	0.002 **	0.006 **	0.008 **

** indicates *p* < 0.01.

**Table 3 biomolecules-13-00231-t003:** *p* values of linear models of intl1 concentrations vs. the relative concentrations of ARGs summed up by antibiotic class and mechanism. The relative concentration of each class and mechanism was log^10^(x*10^6^) transformed except for sulfonamide and protection to meet normal distributions.

			*p* Values
Intl1	Class	Sulfonamide	0.002 **
		Trimethoprim	0.031 *
		Tetracycline	0.689
		Beta-lactamase	0.944
		Multidrug	0.05 *
	Mechanism	Protection	0.003 **
		Deactivate	0.156
		Efflux	0.3

* indicates *p* < 0.05, ** indicates *p* < 0.01.

## Data Availability

Not applicable.

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
