# Peer review of "Antibiotic Resistance Genes in Interconnected Surface Waters as Affected by Agricultural Activities"

_biomolecules, 2023, doi:10.3390/biom13020231_

Round 1

Reviewer 1 Report

The authors discuss the antibiotic resistance genes in interconnected surface waters as affected by agricultural activities. The manuscript delivers some exciting pieces of information and is written in a good manner. With some minor corrections it could be considered for publication:

1.     Rewrite and precise 2.1 section, authors need not explain all details of resistant genes, better write resistance against different classes of antibiotics.

2.     Lines 125 -132, should be a part of the discussion

3.     Authors should have performed a conjugation experiment to study the rate of dissemination of resistant genes among sample sites

4.     Figure A2 and Table A4 from supplementary data should be included in the main paper.

5.     Authors should have detected and confirmed the prevalence of different bacterial strains from each site and data should be added to a separate table

6.     The introduction should be enriched with the latest resistance pattern by adding content and references from the below articles. The information can be added and cited below articles:

https://doi.org/10.3389/fcimb.2019.00193

https://doi.org/10.3389/fmed.2021.677720

https://doi.org/10.3390/pharmaceutics14020295

7.     Entire manuscript requires more language and grammar correction

Reviewer 2 Report

The paper entitled Antibiotic resistance genes in interconnected surface waters as affected by agricultural activities  have a main purpose to assesses the possible impacts of pastures on the prevalence and composition of Antibiotic resistance genes (ARGs) in nearby surface waters, including the ditches around greenhouses and large water bodies. In this frame the water samples were collected from a total of 16 locations, including six ditches around pastures, six ditches around greenhouses and four lakes in the same area. The major goals were the analyse the relative concentration of 29 ARGs that indicate resistance to the four classes of most used antibiotics in the Netherlands livestock and multidrug resisting ARGs, the analyse of relationships between the prevalence of these ARGs, while accounting for the potential influences of horizontal gene transfer and four environmental factors (phosphate, nitrate, pH and DO) on the prevalence of ARGs.

The paper is structured in five parts excepting abstract and references. In part one, Introduction are presented the state of the art into discovery and application of antibiotics in the last century which largely facilitated medical treatments and improved agricultural practices and yields. The authors cited that many of these treatments are currently at risk due to the recent and rapid spreading on a global scale of so-called Antibiotic Resistance Genes (ARGs). Therefore, their rapid spread and occurrence in general and pathogenic bacteria in the environment particularly poses a major problem to medical treatments.

The part two is structured in five chapters in which are detailed the methods used for the obtaining the results according to the goals of the study. In part tree is described in detail the obtained results. This part is detailed in 3 subchapter in which are described and comment the results. Chapter four is dedicated to the discussion of the results into frame of the state of the art in the investigated field.

As conclusions, paper revealed that the ARGs are prevalent in a low-lying agricultural area in the Netherlands connected through waterways. Both pastures and horticulture generate ARG pressures on nearby waterways and impact each other, consequently causing ARG pollution to the environment and may spread ARG risks to humans. The presented results suggest that horizontal gene transfer, next to physical transport, strongly contributes to the widespread distribution of ARGs, particularly so for sulfonamide ARGs.

Please note that the postal address is not complete. 

Reviewer 3 Report

The work entitled “Antibiotic resistance genes in interconnected surface waters as affected by agricultural activities” is a good work. It is necessary some little fix and it is ready to publish.

Figure

1.    Figure 1: the authors should split the histogram in figure 1a based on the ARGs. Similarly figure 1b

2.    Figure 2: add the zero point on graph (for all graphs)

3.    General information: the Authors should use the colours suitable for people with colour-blindness problems, in particular in figure 2. Similar to supplementary figure A2.

General information’s

1.    Check the English

2.    The authors should think to create a paragraph for introduction section for better use of information

Reviewer 4 Report

This paper analyzed 29 ARGs related to the most antibiotics used by livestock in 16 agricultural regions of the Netherlands (representing pasture, greenhouse and lake respectively). The results showed that ARGs showed similar composition in all surface waters around pastures and greenhouses, and sulfa ARGs were dominant. This indicates that both pasture and greenhouse will exert anti biological pressure on adjacent waters. However, lower pressure is found in relatively large and isolated lakes, which indicates that larger water bodies or non-agricultural green buffer zones help reduce the ARG impact in agricultural areas. The content of the article is substantial, the design is reasonable, and the method is properly selected. However, there are some problems that need to be fully reconsidered before the article is published

If the author can provide the geographical zoning map of the sampling location, it will be more helpful for readers to master the spatial information

Environmental factors (pH, dissolved oxygen, etc.) have been mentioned in the method section, but the results section has not been fully displayed, especially the lack of in-depth analysis and presentation of ARG correlation

The author has put too many results in the form of tables in the attachment. A better way might be to put them in the text in the form of graphs, because there are not many results in the text.

The primers used by the authors for ARGs quantification should be provided in the supplementary materials in the form of an attachment table

Round 2

Reviewer 4 Report

Now the manuscript is ready for the publication